

# Semi-supervised learning and bidirectional decoding for effective grammar correction in low-resource scenarios

Zeinab Mahmoud[1], Chunlin Li[1], Marco Zappatore[2], Aiman Solyman[3], Ali Alfatemi[4], Ashraf Osman Ibrahim[5] and Abdelzahir Abdelmaboud[6]

[1] School of Computer Science and Technology, Wuhan University of Technology, Wuhan, Hubei, China
[2] Department of Engineering for Innovation, University of Salento, Lecce, Lecce, Italy
[3] School of Software Engineering, South China University of Technology, Guangzhou, China
[4] Computer Science, Graduate School of Arts and Sciences (GSAS), Fordham University, New York, United States
[5] Advanced Machine Intelligence Research Group, Universiti Malaysia Sabah, Kota Kinabalu, Malaysia
[6] Department of Information Systems, King Khalid University, Muhayel Aseer, Saudi Arabia

Corresponding authors
Chunlin Li, 410144@whut.edu.cn
Aiman Solyman,
aiman_mutasem@hotmail.com

## ABSTRACT

The correction of grammatical errors in natural language processing is a crucial task as it aims to enhance the accuracy and intelligibility of written language. However, developing a grammatical error correction (GEC) framework for low-resource languages presents significant challenges due to the lack of available training data. This article proposes a novel GEC framework for low-resource languages, using Arabic as a case study. To generate more training data, we propose a semi-supervised confusion method called the equal distribution of synthetic errors (EDSE), which generates a wide range of parallel training data. Additionally, this article addresses two limitations of the classical seq2seq GEC model, which are unbalanced outputs due to the unidirectional decoder and exposure bias during inference. To overcome these limitations, we apply a knowledge distillation technique from neural machine translation. This method utilizes two decoders, a forward decoder right-to-left and a backward decoder left-to-right, and measures their agreement using Kullback-Leibler divergence as a regularization term. The experimental results on two benchmarks demonstrate that our proposed framework outperforms the Transformer baseline and two widely used bidirectional decoding techniques, namely asynchronous and synchronous bidirectional decoding. Furthermore, the proposed framework reported the highest F1 score, and generating synthetic data using the equal distribution technique for syntactic errors resulted in a significant improvement in performance. These findings demonstrate the effectiveness of the proposed framework for improving grammatical error correction for low-resource languages, particularly for the Arabic language.

## INTRODUCTION

Automatic correction of grammatical errors is one of the most common NLP tasks in research and industry and it has seen rapid development with the advancement of deep learning techniques. Recent deep neural network approaches are essentially an encoder-decoder architecture (*Solyman et al., 2022*). In GEC neural-based systems, the encoder receives the source, which is an ungrammatical sentence and maps it into an intermediate hidden vector that encodes all the source information. The decoder takes the hidden vector to generate the output correction word by word.

One major challenge in GEC is the lack of available massive parallel training data for low-resource languages, such as Slovenian, Albanian, and Arabic. The classical form of seq2seq GEC often uses a unidirectional decoder that suffers from unbalanced outputs (*Solyman et al., 2022*), which leads the system to generate corrections with good prefixes and bad suffixes. The effects of this problem vary depending on the model structure and the length of the input sequence. However, the autoregressive structure of deep neural network approaches in GEC has a limitation during inference when the previous target word is unavailable; consequently, the model depends on itself and generates a new word that may be out of context, thus generating the so-called exposure bias problem (*Solyman et al., 2022*). The incorrect words generated during inference lead to weakness in the prediction of the next word and result in unsatisfactory correction results. Previous studies such as *Yuan et al. (2019)* sought to use a complementary decoder (R2L) to rerank the n-best list of the L2R decoder, but still the same decoder suffers from an exposure bias problem which leads to bad prefixes corrections.

The current research direction is aimed at lessening the discrepancy that exists between the training and inference stages to increase robustness while feeding erroneous previous predictions to overcome this issue. For instance, a Type-Driven Multi-Turn Corrections approach was proposed by *He et al. (2016)*, which involves constructing multiple training instances from each original instance during training. *Zhang et al. (2018)* proposed a two-stage decoding neural translation model in the inference, that is time-consuming. Another notable work in *Zhang et al. (2019)*, proposed a regularization method during training to increase the agreement between two decoders (L2R and R2L); however, it complicates the training phase because of dynamic sampling and requires more training time and computation resources. To tackle the drawback associated with previous studies, the current work introduces a semi-supervised confusion method that widens synthetic training data. Furthermore, an Arabic grammatical error correction (AraGEC) model was proposed, based on bidirectional knowledge distillation with a regularization method inspired by NMT, as proposed by *Zhang et al. (2022)*, which aims to improve the agreement between the two decoders of forward (R2L) and backward (L2R) into a joint framework. This forces both decoders to act as helper systems for each other and to integrate their advantages to address the exposure bias problem and generate corrections as output with good prefixes and suffixes. The notable outcomes of this work are outlined below:

- A semi-supervised method is proposed to overcome the shortage of parallel training data in AraGEC by generating synthetic training data.
- AraGEC model is proposed based on Transformer-base equipped with a bidirectional knowledge distillation method to address the exposure bias problem typically experienced in automatic GEC systems.
- Experimental results on two benchmarks demonstrate that our model outperforms the current most powerful bidirectional decoding methods as well as previous AraGEC systems.

This article is structured as follows. "Related Work" describes the related works. The proposed confusion method and the GEC framework are presented in "Methodology". "Experiments" examines the experimental details, whereas "Results" reports our evaluation results and analysis. Finally, conclusions are given in "Conclusion and Future Work". The comprehensive set of resources encompasses the codebase, trained models, and essential data files, all of which can be found on GitHub (https://github.com/Zainabobied/SLBDEGC).

## RELATED WORK

Automatic detection and correction of grammatical and other related errors are one of the most popular tasks in NLP, as the interest in it began in the late 1970s with the advent of electronic computing. Rule-based systems were the earliest applications adopted to that end, which use a simple knowledge base that contained all the grammar rules of the relevant language (*Simmons, 1978*). In the 1990s, there was a significant development in the field of computational linguistics that led to the use of n-gram language models to measure the probability of characters and words in a contiguous sequence from a given sample of text (*Brown et al., 1992*). Recently, GEC can be considered a machine translation task, which translates text with errors (interpreted as the source language) into error-free text. GEC-based SMT is a phrase-based system that optimizes the conditional probability of finding the correct sentence $Y$ given the input sentence $X$, among all possible corrections (*Junczys-Dowmunt & Grundkiewicz, 2016*). Due to the increases in computer processing capabilities and the availability of massive training data, GEC-based NMT systems demonstrated the ability to outperform the previous and more traditional GEC techniques thanks to their new approach that allows to correct texts using a set of hidden layers in the form of seq2seq models such as Recurrent Neural Network (RNN), Convolutional Neural Network (CNN), or Transformer (*Solyman et al., 2022*).

Extensive research attention has been devoted to the English and Chinese languages, primarily due to the availability of rich resources including parallel text corpora, pre-trained models, and open-access GEC systems. Notably, GPT-3, a pre-trained language model with 175 billion parameters, has demonstrated powerful performance in generating natural language text, particularly in the field of English GEC (*Brown et al., 2020*). Google AI unveiled Pathways, a mega language model with a staggering capacity of 540 billion parameters. This model has achieved a remarkable level of human-like performance in a wide range of numerous NLP tasks, including GEC (*Chowdhery et al., 2022*).
[1] The term 'low-resource languages' in NLP encompasses languages with limited resources, including training data, not solely confined to GEC. While GEC is affected by low-resource languages, the concept extends to other areas within the field of NLP.

However, the main challenge of low-resource languages[1] such as Italian, French, and Arabic is the lack of such resources. *Ge, Wei & Zhou (2018)* proposed an approach called *fluency boost learning* based on CNN. This iterative routing process effectively corrects texts and leads to a substantial improvement in the accuracy and fluency of GEC systems. In another study, *Acheampong & Tian (2021)* introduced a GEC system that minimizes the reliance on extensive training data, a common requirement for neural-based GEC systems; the system used cascaded learning strategies. In their study, *Wan, Wan & Wang (2020)* introduced a data augmentation method with the objective of improving the performance and robustness of GEC models. This approach centers around modifying latent representations of grammatical sentences to generate synthetic training samples encompassing various grammatical errors. A copy-augmented approach based on the Transformer baseline in Indonesian GEC was proposed by (*Zhao et al., 2019*). This method enhances accuracy by copying correct or unchanged words from the source text into the target text, resulting in improved performance. *Sun et al. (2022)* introduced a unified strategy to enhance multilingual GEC models. Their approach involved leveraging a pre-trained cross-lingual language model and employing synthetic data construction techniques. By utilizing the non-autoregressive translation capability of the pre-trained model, a diverse set of error-corrected data was generated, which was subsequently employed for pre-training the GEC models. *Náplava & Straka (2019)* introduced a GEC based on a neural machine translation approach, and also proposed a spell-confusion method that was utilized to generate synthetic corpora from clean monolingual data that was introduced specifically designed for GEC in Czech. The GitHub Typo *Corpus*, introduced by *Hagiwara & Mita (2020)*, serves as a valuable resource for training GEC models. This vast multilingual training dataset comprises 15 languages.

AraGEC has gained significant attention due to its successful performance in shared tasks during 2014 and 2015, as reported by *Mohit et al. (2014)* and (*Rozovskaya et al., 2015*). Although AraGEC was recognized early on, it faces a significant challenge due to the limited availability of training data. The existing annotated Arabic training data consists of only 20,430 examples. To tackle this issue, *Rozovskaya et al. (2014)* introduced Columbia a GEC system that achieved the top ranking in the QALB-2014 Shared Task on Arabic GEC. This system incorporates statistical models, linguistic resources, and rule-based modules designed to tackle various types of errors. *Nawar (2015)* proposed a framework characterized as a probabilistic rule-based system that extracts error correction rules and assigns probabilities to each rule. The proposed framework achieved the highest $F_1$ score in the QALB-2015 Shared Task. *Sina (2017)* utilized an attention-based RNN encoder-decoder model as the first end-to-end neural-based AraGEC system. *Watson, Zalmout & Habash (2018)* employed seq2seq Bidirectional Recurrent Neural Networks (BRNN) and FastText word embeddings to extract richer linguistic information for their model. *Solyman, Wang & Tao (2019)* introduced a convolutional AraGEC model, which was later extended in their subsequent work (*Solyman et al., 2021*). This extended GEC model incorporates a spell-confusion approach and a CNN seq2seq model with shared embedding and fine-tuning, enhancing its capabilities for accurate error correction. In another work by *Solyman et al. (2023)*, seven data augmentation approaches were

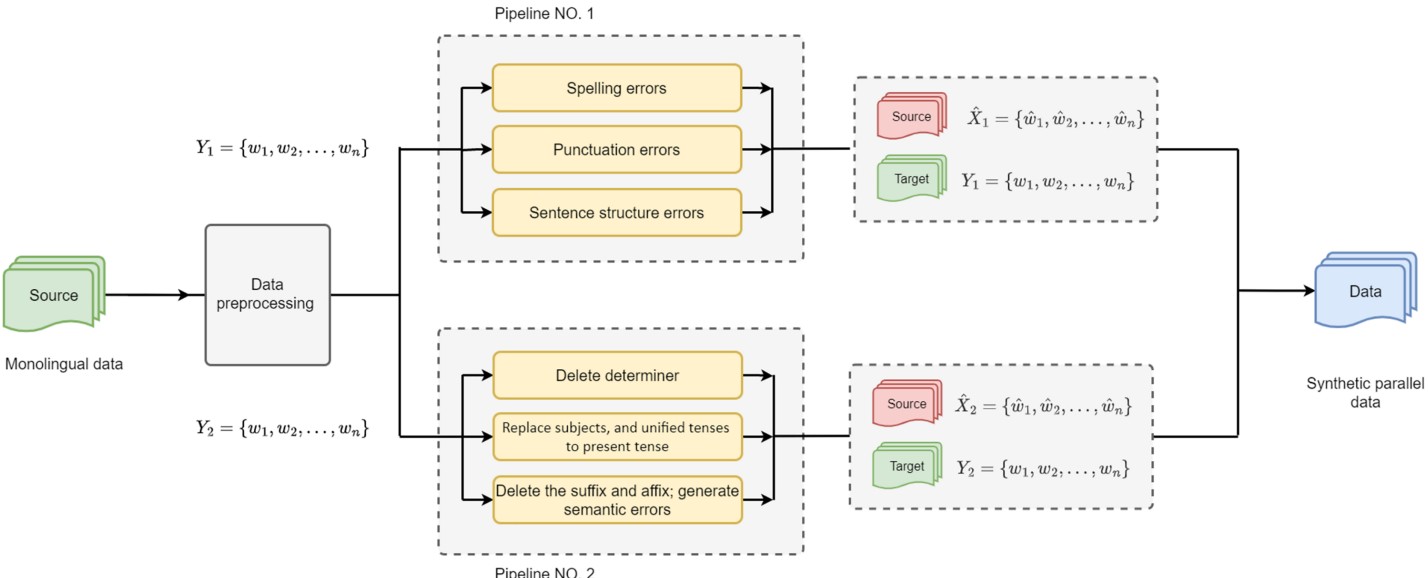

**Figure 1** Architecture of the Equal Distribution of Synthetic Errors (EDSE) approach is made of two synthetic pipelines that have the same probability of error generations, green refers to the original data, red is the synthetic data (erroneous), and blue is the parallel training data.

proposed to increase the impact of source contribution in GEC systems. These approaches led to improvements without the need for additional training data. *Pajak & Pajak (2022)* investigate the application of multilingual sequence-to-sequence models for GEC. The research shows that using a single model to address error correction in multiple languages is effective. To achieve this, large pre-trained models such as mBART, mT5, and ProphetNet were tuned. The study finds that fine-tuning large models for GEC can be done with limited computational resources, and emphasizes that the size of the model plays a crucial role in determining the quality of the results.

As it can be inferred from the in-domain literature overview provided so far, the existing systems for low-resource scenarios predominantly use spell-confusion methods to generate synthetic data that almost lacks diversity, thus leading to limited training patterns and, consequently, limiting significantly the true application potential of those systems. Therefore, an extended effort is needed to introduce more efficient approaches capable of addressing the lack of training data and the exposure bias problem.

## METHODOLOGY

### System overview

In this section, we introduce the proposed GEC framework in detail and formulate the hypotheses. Initially, a novel approach was proposed to construct reliable synthetic parallel training data for GEC, as shown in Fig. 1. Furthermore, we introduce a knowledge distillation with bidirectional decoding for AraGEC based on Transformer. This technique was proposed by *Zhang et al. (2022)* in NMT, and we have successfully integrated it into our model.

## Noise method

Despite the widespread use of Arabic on the Internet, there is still a lack of freely available training data for NLP applications such as semantic analysis (*Baghdadi et al., 2022*), text classifications (*Masri & Al-Jabi, 2023*), and automatic grammar correction. The Qatar Arabic Language Bank (http://nlp.qatar.cmu.edu/qalb/) (QALB) is the only available annotated parallel data for GEC: it consists of 20,430 examples, which is not enough to train GEC neural-based systems effectively. Furthermore, building extensive parallel training data for GEC is expensive, time-consuming, and requires appropriate tools. To this end, numerous methods have been proposed such as back-translation (*Kiyono et al., 2020*) and misspelling confusion sets (*Grundkiewicz, Junczys-Dowmunt & Heafield, 2019*) to overcome the lack of training data. However, these techniques are unreliable to construct high-quality training data containing the most common grammatical errors (training patterns) and cannot control the types of errors, rate, and distribution.

While GEC has gained research attention in recent years, there has been a neglect of error identification and classification, particularly in the context of AraGEC. Previous approaches have focused on classifying grammatical errors in English, including style errors, spelling errors, semantic errors, syntax errors, missing words, extra words, and agreement errors (*Naber, 2003*; *Wagner, Foster & van Genabith, 2007*; *Yuan, 2017*). In the case of Arabic, *Zaghouani et al. (2014)* have classified seven error types, such as spelling, punctuation, incorrect word choice, morphology, syntactic, proper name errors, and dialectal usage correction, which were utilized in Arabic GEC shared tasks (*Mohit et al., 2014*; *Rozovskaya et al., 2015*). Drawing from the existing literature, our proposed method considers five main types and 14 sub-types of errors for Arabic, as outlined in Fig. 2, which includes examples and translations. These error types serve as the foundation for our approach, ensuring comprehensive coverage of the most common errors in the Arabic language, even those that were not addressed in *Zaghouani et al. (2014)* study.

The seed of our synthetic data was a monolingual *corpus* namely CC100-Arabic, created by *Conneau et al. (2020)* from Facebook AI. The data was collected during January–December 2018 Commoncrawl snapshots from the CC-Net repository, and the total data size was 5.4 GB organized into a single text document. The CC100 arabic *corpus* was selected because it is freely available and it is the most extensive monolingual Arabic *corpus*. In addition, it contains various topics such as education, history, economy, law, health, stories, cooking recipes, and sport. Several steps of data prepossessing were initially applied over the given *corpus*, such as removing the duplicate paragraphs and spaces between lines. We decided to use 25 million examples in different lengths, between 10 to 100 words. Then, data was normalized from diacritical marks, non-UTF8 encoding, links, and mentions, and we kept punctuation, numbers, and Arabic stop words.

The performance of GEC systems was improved thanks to the monolingual data, which was used during training to provide more training patterns; this depends on the size and quality of the synthetic data (*Grundkiewicz, Junczys-Dowmunt & Heafield, 2019*). This article proposes a semi-supervised method for generating massive synthetic data that contains the most common types of grammatical errors in Arabic. In order to cover a wide

| Error class | Error type | Example | Translation | EDR* |
|---|---|---|---|---|
| Spelling | Dialectal word | إنه أطول كوبري علي نهر النيل | It is the longest arch over the Nile River | High |
| | Typographical | إنه أطول جسرر علي نهر النيل | It is the longest bridgge over the Nile River | High |
| | Special characters | انه اطول جسر على نهر النيل | There are no corresponding special characters in English such as أ، ي، آ. | High |
| | Proper name | دانياكوشان أطول جسر في العالم | Danyakushan is the longest bridge in the world. | Low |
| Syntax | Verb-subject agreement | قرأت الطالب الكتاب | The student reads the books | Middle |
| | Noun determiners | الإجتهاد أساس نجاح | Hard word is foundation - success | High |
| | Verb tense | تدرس بإجتهاد ونجحت | She is study hard and succeed | Low |
| | Preposition | يدرس الطالب الي المكتبة | The student study to the school | Middle |
| | Morphological | الطالبتان يجتهدوا لتحصيل النجاح | The two students tries hard to achieve successes | Middle |
| Semantic | Contextual | السباحة أساس النجاح | Swimming is the key to success | Low |
| | Wrong word choice | الإجتهاد أنيس النجاح | Hard work is the friend the success. | Low |
| Sentence structure | Fragment | نجح الطالب إذ إجتهد | The student will succeed . If he worked hard | Middle |
| | Run-ons | الطالبتان يدرسن هناو يدرسن هناك | The two students study here. They also study there. | Low |
| Punctuation | Punctuation | الإجتهاد أساس، النجاح | Hard work is the key, to success | Middle |

**Figure 2 Examples of error classes in Arabic intended for generation, including error types (sub-classes), illustrative examples (with incorrect words highlighted in red), translations, and error distribution rate (EDR) for each error type.**

range of errors in AraGEC, two pipelines were applied; hence the type of errors was grouped into two groups: group one includes spelling errors, sentence structure, and punctuation errors; while group two includes syntax and semantic errors, as shown in Fig. 2, which include examples, translation, and the rate. We relied on the error distribution rates in the Arabic learner *corpus* (ALC) (*Alfaifi, Atwell & Hedaya, 2014*), in which punctuation and spelling have high rates, respectively. Next are the syntax errors (*middle*), semantic and sentence structure errors that have the lower rate (*low*), respectively. This makes it easy to control the rate and distribution of each type of error.

The proposed method has two key parameters: $N$ refers to a rate of words to be processed and has an initial value between 0 and 1, we set the value of $N$ during training to 0.1; $T$ is the total number of words in each input sentence. Let us begin with the first pipeline, which generates errors that include *Misspelling* (with four sub-classes), *Punctuation*, and *Sentence structure* (with two sub-classes). For example, in the generation of spelling errors, one method aims to tokenize the input sentence and then randomly selects a word to either delete a character or add more characters. Punctuation errors are

introduced by injecting specific punctuation marks from a given list or by removing existing punctuation. Sentence structure errors are induced through a method that transforms the input sentence into a part-of-speech (PoS) tagging format, followed by one of these two operations: (1) Swapping two sentence components: the subject, object, or verb, within the standard word order of VSO (verb-subject-object) in Arabic. For example, it may entail reordering the sentence from VSO to SVO (subject-verb-object) or from VSO to OVS (object-verb-subject); or (2) removing one of the sentence structures.

The second pipeline in the proposed method tackles the most complex error types: *Syntax* (with five sub-classes) and *Semantic* (with two sub-classes). To begin, each input sentence is initially converted into the PoS format. Within this pipeline, a total of seven operations (functions) are applied at each iteration $t$. These operations include but are not limited to, the following examples: (1) deletion of determiners; (2) replacement of the subject with another word from the *corpus* vocabulary, resulting in a verb-subject disagreement; (3) unification of the tense to the present tense format by utilizing PoS tags, disregarding future and past tenses, thus introducing tense verb errors; (4) deletion of suffixes and affixes to create morphological errors and inconsistencies within the sentence; (5) substitution of a random word in the sentence with a word from the dataset, leading to semantic errors that confuse the reader and impact the sentence context. The proposed method is named Equal Distribution of Syntactic Errors (EDSE), and its architecture is depicted in Fig. 1. Algorithm 1 provides an overview of the procedures and data flow, illustrating a simplified representation with only six functions, whereas the actual implementation encompasses a total of 16 functions. Each main error class, such as *Misspelling*, *Punctuation*, and *Syntax*, consists of multiple sub-classes. During each iteration $t$ consisting of three steps, a random sub-function is chosen from each main class, forming a pipeline that incorporates the selected sub-functions. However, the proposed method is not perfect and there is room for refinement, it serves as a starting point for generating different types of errors in Arabic text.

## Bidirectional decoding

The proposed AraGEC framework utilizes both forward and backward decoders in its decoding structure. Since Arabic is a right-to-left (R2L) language, the decoder that moves in a forward direction employs a mask matrix in the form of an upper triangular shape, allowing it to consider the information on the right of $y_t$. This forward decoder is referred to as the R2L decoder. The backward decoder in the regular language model perceives the sequences from left to right and is named the L2R decoder. Furthermore, a lower triangular mask matrix was used in the L2R decoder. Both given decoders are utilized to detect and correct the next token, where $t$ represents the current token index ranging from 1 to $T$ in the sequence. The decoders operate on the range of tokens from ($t + 1$ to $T$) or (1 to $t - 1$) given the source $X$ and the target $Y$, as described by the following equations.

$$logP(y|\mathbf{X};\overleftarrow{\theta}) = \prod_{n=1}^{N} P(y_t|y_{t+1:T}, \mathbf{X}; \overleftarrow{\boldsymbol{\theta}}), \quad (1)$$

---

**Algorithm 1   Equal distribution of synthetic errors.**

**Require:** $Y_1, Y_2, \alpha$.      ▷ Pair of original monolingual sentences, $\alpha$ value between 0–1

**Ensure:** $(\hat{X}_1, Y_1), (\hat{X}_2, Y_2)$.      ▷ Pair of parallel synthetic examples

    **function** GENERATECHARACTERERROR $(\hat{X}_i)$

        $\hat{X}_i = [w_1, w_2, ..., w_n], w_i = [c_1, c_2, ..., c_n], c_i \in [c_1, c_2, ..., c_n]$      ▷ Choose a character $c_i$

        $\hat{c}_i \in [c_1, c_2, ..., c_n]$      ▷ Delete $c_i$ or Add $\hat{c}_i$ to position $i + 1$

        $\hat{w}_i = [c_1, c_2, c_i, \hat{c}_i, ..., c_n] \hat{X}_i = [w_1, w_2, ..., w_n]$      ▷ Update $\hat{w}_i$ and $\hat{Y}_i$

        **return** $\hat{X}_i$

    **end function**

    **function** GENERATEPUNCTUATIONERROR $(\hat{X}_i)$

        $\hat{X}_i = [w_1, w_2, ..., w_n], lst = [!, '', (, ), *, ., :, ?, [, ], \{, \}]$      ▷ $lst$ is a list of punctuation

        $\hat{X}_i = [w_1, w_2, lst_i, ..., w_n]$      ▷ Insert $lst_i$ in a random position within $\hat{X}_i$

        **return** $\hat{X}_i$

    **end function**

    **function** GENERATESTRUCTURALERROR $(\hat{X}_i)$

        $\hat{X}_i = [w_1, w_2, ..., w_n]$      ▷ Convert $\hat{X}_i$ to PoS tags

        $\hat{X}_i = [w_1, w_n, ..., w_2]$      ▷ Delete or swap sentence components [subject, object, verb]

        **return** $\hat{X}_i$

    **end function**

    **function** GENERATEDETERMINERERROR $(\hat{X}_i)$

        $\hat{X}_i = [w_1, w_2, ..., w_n]$      ▷ Convert $\hat{X}_i$ to PoS tags

        $\hat{X}_i = [w_1, -, ..., w_n]$      ▷ Delete a single element of determiner, suffix, prefix, etc.

        **return** $\hat{X}_i$

    **end function**

    **function** GENERATEVERBSUBJECTERROR $(\hat{X}_i)$

        $\hat{X}_i = [w_1, w_2, ..., w_n], lst = [\hat{w}_1, \hat{w}_2, ..., \hat{w}_n]$      ▷ $lst$ is a list of training vocabulary words

        $\hat{X}_i = [w_1, w_2, \hat{w}_i, ..., w_n]$      ▷ Replace a subject with $\hat{w}_i$ from the list

        **return** $\hat{X}_i$

    **end function**

    **function** GENERATESEMANTICERROR $(\hat{X}_i)$

        $\hat{X}_i = [w_1, w_2, ..., w_n], lst = [\hat{w}_1, \hat{w}_2, ..., \hat{w}_n]$      ▷ $lst$ is a list of training vocabulary words

        $\hat{X}_i = [w_1, w_2, \hat{w}_i, ..., \hat{w}_j]$      ▷ Replace 2 elements from $\hat{X}_i$ with 2 random entries from $lst$

        **return** $\hat{X}_i$

    **end function**

    **procedure** EQUALDISTRIBUTIONSYNTHETICERRORS $(Y_1, Y_2, \alpha)$

        $\hat{X}_1 \leftarrow Y_1, N1 \leftarrow (\alpha \times len(Y_1))$

        $fns = [GenerateCharacterError, GeneratePunctuationError, GenerateStructuralError]$

        **for** $i = 1$ to $N1$ **do**

---

| **Algorithm 1** (continued) | |
|---|---|
| $\widehat{X}_i \leftarrow \text{choice}(fns)(\widehat{X}_i)$ | ▷ Apply one of three functions in *fns* |
| **end for** | |
| $X_2 \leftarrow Y_2, N2 \leftarrow (\alpha \times \text{len}(Y_2))$ | |
| $fnx = [\text{GenerateDeterminerError}, \text{GenerateVerbSubjectError}, \text{GenerateSemanticError}]$ | |
| **for** $i = 1$ to $N2$ **do** | |
| $\widehat{X}_i \leftarrow \text{choice}(fnx)(\widehat{X}_i)$ | ▷ Apply one of three functions in *fnx* |
| **end for** | |
| **return** $(\widehat{X}_1, Y_1), (\widehat{X}_2, Y_2)$ | |
| **end procedure** | |

$$logP(y|\mathbf{X}; \vec{\theta}) = \prod_{n=1}^{N} P(y_t|y_{1:t-1}, \mathbf{X}; \vec{\boldsymbol{\theta}}). \qquad (2)$$

The literature of the previous work in AraGEC demonstrates that the R2L performs better than the L2R decoder as described by *Solyman et al. (2022)*; hence, in this work the backward decoder (R2L) will be the student and the forward decoder (L2R) represent the teacher. R2L decoder learns dependencies of the output sequences from right to left, whereas the L2R learns the dependencies of the output sequences from left to right, and this is the relative future information of the R2L. Thereon, the output of both decoders (R2L—L2R) which is the probability distribution of words in each position that can be represented as complementary information of two decoding sides. This makes the model force the probability distribution of $P_{R2L}$ and $P_{L2R}$ to support each other during training to generate future information, as shown in the following equation.

$$P_{R2L}(y_t = w|y_{1:t-1}, \mathbf{X}) \sim P_{L2R}(y_t = w|y_{t+1:T}, \mathbf{X}) \qquad (3)$$

where $w$ is the given token from the training vocabulary, and $t$ refers the $t_{th}$ position of the output corrected sequence. However, these decoders cannot improve equally and cannot fulfill Eq. (4) if optimized separately using the standard MLE. The L2R decoder cannot learn the global coherence from R2L and this will lead to unsatisfactory corrections. To this end, a knowledge distillation method was proposed to improve both decoders during training process and the transferred information learning across R2L and L2R decoders. Furthermore, the L2R decoder will not be used during inference so as to not affect decoding speed as compared to the conventional GEC models that used the L2R model during inference.

### Knowledge distillation
The main objective of the proposed knowledge Distillation method is to incorporate the learning information from the backward decoder to the forward decoder, which uses L2R decoder as a teacher that has future knowledge (hidden states) of R2L decoder. This approach utilizes the logits and the teacher's final layer hidden states model for increased

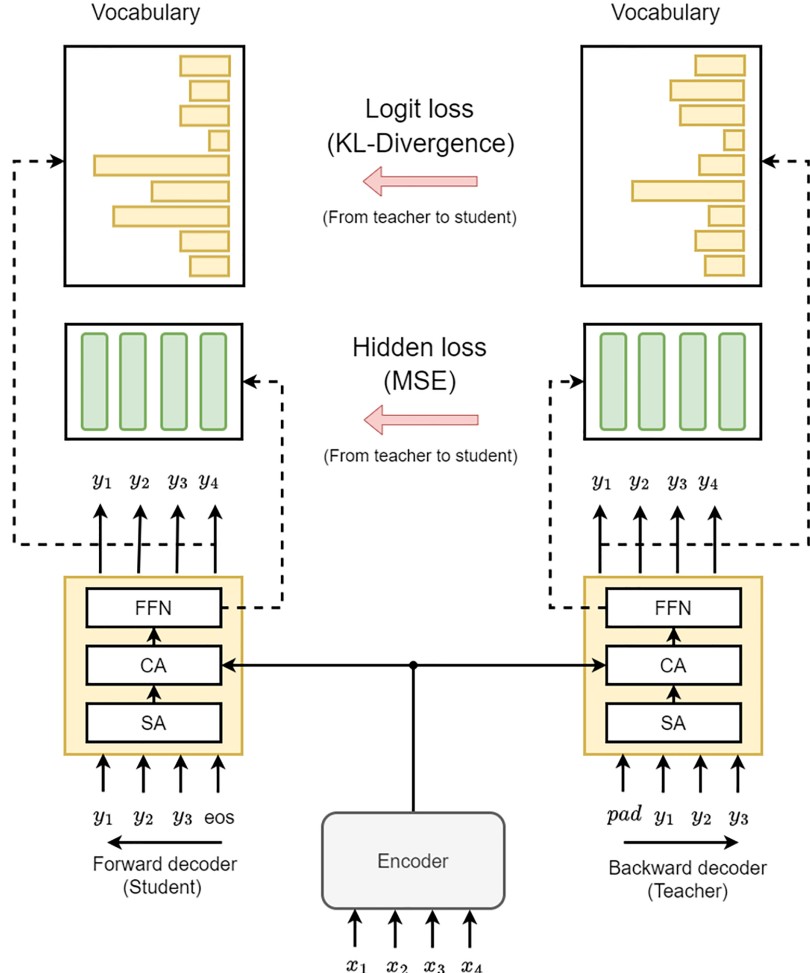

**Figure 3 The design of our BKDGEC model incorporates two decoders, labeled as Backward and Forward, represented by yellow boxes.** These decoders consist of Self-Attention (SA), Cross-Attention (CA), and a Feed-Forward Neural Network (FFN).

versatility and effectiveness. Furthermore, since the student and teacher models will learn during training at the same time, so we called this method Bidirectional Knowledge Distillation Grammatical Error Correction (BKDGEC), which encompasses hidden state-based distillation and logit-based as depicted in Fig. 3. In this context, we would like to clarify that the term "knowledge distillation" typically implies transferring information from a larger model to a smaller model. However, in our framework, we use the term "knowledge distillation" to describe the transfer of knowledge from the teacher to the student decoder, even though they have similar sizes.

In the realm of neural-based techniques, Logit alludes to the predictive vector that can be produced using the last layer of the decoder. This layer has the same dimension as the vocabulary size and is employed to determine the token that should be predicted in the present time step. In this work, Kullback-Leibler (KL) divergence was utilized to quantify the divergence between the logit probability distributions of the backward and forward

decoders at the same position. Eqs. (4) and (5) demonstrate the implementation of this method:

$$L_{logit} = \sum_{n=1}^{T} KL(P(y_t|y_{1:t-1}, \mathbf{X}\vec{\theta})||P(y_t|y_{t+1:T}, \mathbf{X}; \bar{\theta})), \tag{4}$$

$$KL(P(y_t|y_{1:t-1}, X\overrightarrow{\theta})||P(y_t|y_{t+1:T}, X; \overleftarrow{\theta}))$$
$$= \sum_{w \in V} P(y_t = w|y_{1:t-1}, \mathbf{X}; \overrightarrow{\boldsymbol{\theta}})) \times \log \frac{\mathbf{P}(\mathbf{y_t} = \mathbf{w}|\mathbf{y_{1:t-1}}, \mathbf{X}; \overrightarrow{\boldsymbol{\theta}})}{\mathbf{P}(\mathbf{y_t} = \mathbf{w}|\mathbf{y_{t+1:T}}, \mathbf{X}; \overleftarrow{\boldsymbol{\theta}})}. \tag{5}$$

Here, $V$ represents the output vocabulary, and $T$ denotes the target length. Consequently, this led to the distillation of hidden states, which can be depicted through the following equation.

$$L_{hd} = MSE(\overleftarrow{H}W_h, \vec{H}) \tag{6}$$

where $MSE$ is a loss function stands to mean squared error, $\overleftarrow{H} \in R^{l \times \acute{d}}$ and $\overrightarrow{H} \in R^{l \times d}$ refers to the hidden states of the both decoders R2L and L2R, respectively. Furthermore, $W_h \in R^{\acute{d} \times d}$ is a linear function that adjusts the L2R hidden states to have the same dimension as the R2L hidden states, and $\acute{d}$ $d$ are the hidden dimension of both the decoders and have the same value. In this work, two knowledge distillation functions were utilized to encourage the backward decoder to grasp future representations. In addition, a joint training framework was constructed to optimize both the decoders iteratively, as shown in Eq. (7).

$$L(\theta) = \sum -\log P(\vec{y}|X, \vec{\theta}) - \log P(\bar{y}|X, \bar{\theta}) + L_{kd}(\vec{y}, \bar{y}), \tag{7}$$
$$L_{kd} = L_{logit} + L_{hd}, \tag{8}$$

As explained, the knowledge distillation learning process in this work is based on a student model imitating the teacher model. This might raise concerns as the student's potential might be constrained by the teacher's performance, resulting in limited ability to surpass the teacher (*Clark et al., 2019*). Consequently, the student model could rely heavily or excessively on the teacher model. To tackle this challenge in our BKDGEC framework, we applied two distillation methods. These methods help the R2L decoder gain a better understanding of future knowledge and drive the model to place more emphasis on the L2R decoder as training progresses. To this end, an annealing mechanism was proposed that is fitting for BKDGEC. It adjusts the training objective to consider the agreement between both decoders as in Eq. (9).

$$L(\theta) = \sum_{i=1}^{n} \left[ -(1-\lambda) \cdot \left( \log P_{\bar{\theta}}(y_i|X_i) \right)^2 - \lambda \cdot \log P_{\vec{\theta}}(y_i|X_i) + (1-\lambda)\lambda \cdot L_{kd}(y_i, \hat{y}_i) \right], \tag{9}$$

where $\lambda \in [0, 1]$ is a hyperparameter that controls the balance between the forward decoder $P_{\vec{\theta}}$ and the backward decoder $P_{\bar{\theta}}$. Here, $y_i$ is the ground truth label for the $i$-th input sample $X_i$, and $\hat{y}_i$ is the output label from the forward decoder. The value of $\lambda$ is

determined based on the current training step $c_{step}$ and the warm start step $w_{step}$. Specifically, if $c_{step} < w_{step}$, then $\lambda = 1$, and the training objective function only considers the output of the forward decoder $P_{\vec{\theta}}$ to help the backward decoder $P_{\overleftarrow{\theta}}$ acquire sufficient knowledge. Otherwise, $\lambda = \frac{w_{step}}{c_{step}}$, indicating that the number of training steps is greater than $w_{step}$. In this case, the effect of the backward decoder $P_{\overleftarrow{\theta}}$ (also known as the teacher) increases, and the initial value of the divergence in agreement $L_{kd}(y_i, \widehat{y}_i)$ also increases during training, while the output of the forward decoder $P_{\vec{\theta}}$ (also known as the student) decreases over time.

## EXPERIMENTS

### Data

The synthetic parallel training data used in our study originated from CC100-Arabic, a dataset introduced by *Conneau et al. (2020)* from Facebook AI. Prior to utilization, the data underwent preprocessing, and subsequently, our proposed confusion method, EDSE, was applied to generate parallel training data, which was then divided into training and development sets. For fine-tuning purposes, we used the data from QALB-2014. This dataset provided us with 20,430 examples, making it a valuable resource for improving the performance of our AraGEC system. The QALB-2014 *corpus* consisted of English articles that had been translated into Arabic using machine translator, as well as the Arabic Learners Written *Corpus* (ALWC) introduced by *Alfaifi, Atwell & Hedaya (2014)*. Additionally, it included users' comments from the Al-Jazeera news platform that contained a range of grammatical errors due to the various Arabic dialects represented. A team of linguistic experts and native speakers corrected and annotated the data to ensure the accuracy and reliability of the target sentences.

### Model setting

The initial baseline model used in our experiments was based on the Transformer architecture, as introduced by (*Vaswani et al., 2017*). However, certain modifications were made to adapt it to our specific task. The model size was reduced from 512 to 256, and the batch size was reduced from 2,048 to 128. These adjustments were necessary because our proposed model operated on chunks of 2-to-4 characters instead of whole words, which required smaller dimensions to ensure optimal performance. Additionally, the number of layers in the Transformer was reduced from six to four during our experiments, while the number of attention heads remained at eight, consistent with the original model. In the encoder and decoder, the first layers were used for positional encoding instead of the static encoding method employed in BERT (*Devlin et al., 2019*). However, we did not apply label smoothing in our approach. To tackle overfitting, we employed the Adam optimizer (*Kingma & Ba, 2015*) instead of using warm-up and cool-down steps for adjusting the learning rate. We set the learning rate to 0.003 during the training phase and reduced it to 0.001 during the fine-tuning phase. Furthermore, to prevent the gradient from exploding, we implemented gradient clipping with a threshold value of 1.0. Dropout regularization was also utilized with probabilities of 0.15 and 0.10 during training and fine-tuning, respectively. To handle the challenge of rare words, we applied the Byte Pair Encoding

(BPE) algorithm proposed by *Sennrich, Haddow & Birch (2016)*. This algorithm enables the segmentation of unknown tokens into sub-tokens, improving the model's ability to handle out-of-vocabulary words and enhance overall performance.

Early stop was applied during training, which led to 27 epochs using the monolingual parallel synthetic data and three epochs for fine-tuning using parallel authentic data of QALB-2014. A checkpoint of the best model was created after each epoch. Due to the small chunks of input sequences, the maximum length of input sequences was set to 400 tokens in training and testing. The tokenizer was the BPE algorithm with 1,000 vocabulary size. Beam search was applied during inference with a five-beam size. The outputs of the test set have been tuned after inference using a simple data preprocessing method to remove the repetitions of words, characters, and some punctuation errors that the model failed to correct well.

In this context, we acknowledge that the hyperparameters were deliberately set manually. The decision to manually select hyperparameters was made to ensure careful consideration of the unique characteristics of the data and task at hand. We acknowledge that automated or systematic techniques for hyperparameter selection exist, but due to the constraints of our study and the need for fine-tuned control over the models, we opted for a manual approach. This allowed us to tailor the hyperparameter values to our specific context and strike a balance between model performance and generalization.

### Evaluation

We evaluated the proposed framework using two benchmarks: QALB-2014 and QALB-2015, which were used for the Arabic GEC shared tasks in *Mohit et al. (2014)*, (*Rozovskaya et al., 2015*). To assess the performance, we applied MaxMatch and the same tool used in the aforementioned shared tasks. This tool measured the word-level edits in the output compared to the reference sentences and provided precision, recall, and $F_1$ scores under different training scenarios. Furthermore, we applied the BLEU-4 score to evaluate the quality of the machine-corrected sentences compared to high-quality human-corrected sentences.

## RESULTS

This section investigates the performance of the proposed framework, including the impact of the synthetic data, the bidirectional knowledge distillation method, as well as fine-tuning and re-ranking L2R as an improvement. We also investigated the performance against the most powerful bidirectional methods in NMT: asynchronous and synchronous decoding.

### Impact of synthetic data

The constructed synthetic data have more diverse examples than those used to train BKDGEC. Table 1 shows the effectiveness of our EDSE method to construct more reliable data compared to previous approaches such as a semi-supervised confusion function (SCF) (*Solyman et al., 2021*) and a simple spelling noise method (SSNM) (*Solyman et al., 2022*) using the same data size for a fair comparison consisting of 250 k examples from each

**Table 1 Performance of asynchronous and synchronous decoding in AraGEC using the same baseline (Transformer) compared to BKDGEC.** Bold indicates the highest scores.

| Training data | QALB-2014 | | | | QALB-2015 | |
|---|---|---|---|---|---|---|
| | Prec. | Recall | $F_1$ | Prec. | Recall | $F_1$ |
| SCF | 59.11 | 36.41 | 45.06 | 61.91 | 39.78 | 48.43 |
| SSNM | 62.01 | 39.51 | 48.26 | 63.17 | 42.76 | 50.99 |
| EDSE | **63.23** | **42.01** | **50.48** | **64.37** | **45.51** | **53.32** |

training set. The three synthetic sets have been used to train the baseline Transformer base without fine-tuning, BPE was applied with a vocabulary of 30k to reduce the confusion caused by unknown words during training. EDSE performed better than SCF and SSNM in the benchmark QALB-2014. This highlights the importance of multi-training patterns in the training data, in which SCF contains only spelling errors, while SSNM has more training patterns but is still limited compared to our synthetic data.

Eventually, the performance was investigated using the full synthetic data for training the model. Table 2 shows that F1 score increased +18.71 and +3.06 for QALB-2014 and QALB-2015, respectively. This emphasizes the importance and ability of producing synthetic data to raise the level and effectiveness of the GEC systems during training, and also the impact of 10k vocabulary of the BPE algorithm.

## Impact of bid-knowledge distillation

The performance of different versions of the proposed GEC framework, utilizing two benchmarks, is presented in Table 2. The baseline model was a Transformer-based approach trained on the QALB-2014 authentic *corpus*, with slight modifications. The results demonstrate that the proposed BKDGEC regularization technique can significantly enhance the framework's performance, as indicated by the $F_1$ scores of 69.73 and 72.08 for QALB-2014 and QALB-2015, respectively. Notably, the bid-knowledge distillation approach proved to be particularly effective in improving the framework's performance, highlighting the backward decoder's ability to predict the forward decoder's concurrent states accurately. These findings have significant implications for the development of more effective GEC frameworks.

## Impact of fine-tuning

BKDGEC has been carefully fine-tuned to improve its accuracy and performance. This fine-tuning process involved using the original parallel *corpus* of QALB-2014 and a monolingual dataset called CC-100 (https://data.statmt.org/cc-100/), consisting of 1 k clean sentences. The results of this process were presented in Table 2, which achieved the best results among all models with an F1 score of 70.29% for QALB-2014 and 73.13% for QALB-2015. The impact of fine-tuning on both datasets was remarkable, as demonstrated by the significant improvement in the model's accuracy. Notably, the parallel *corpus* yielded better results, likely due to the inclusion of additional authentic examples. These

**Table 2 Comparisons of precision, recall, and $F_1$ of the baseline, with EDSE data, bidirectional knowledge distillation method (BKDGEC), fine-tuning, as well as L2R re-ranking.** Bold indicates the highest scores.

| Model | QALB-2014 | | | | QALB-2015 | |
|---|---|---|---|---|---|---|
| | Prec. | Recall | $F_1$ | Prec. | Recall | $F_1$ |
| Transformer (baseline) | 75.61 | 55.82 | 64.22 | 74.78 | 60.86 | 67.10 |
| Transformer + EDSE data | 77.14 | 62.73 | 69.19 | 75.36 | 67.53 | 71.23 |
| Transformer + EDSE data + BKDGEC | 77.91 | 63.11 | 69.73 | 76.17 | 68.42 | 72.08 |
| Transformer + EDSE data + BKDGEC + fine-tuning | 78.12 | 63.90 | 70.29 | 76.89 | 69.73 | 73.13 |
| Transformer + EDSE data + BKDGEC + fine-tuning + L2R re-ranking | **78.61** | **65.59** | **71.51** | **78.21** | **70.28** | **74.03** |

findings highlight the importance of using high-quality datasets for fine-tuning language models, as it can have a significant impact on GEC performance.

## Re-ranking n-best list

We applied re-ranking from NMT to enhance the performance after inference, which achieved significant improvement (*Liu et al., 2016*). Initially, three different models were trained on both sides (R2L and L2R) using BKDGEC method from scratch which utilized the synthetic data for training and which was tuned using QALB-2014. This enriches the hypothesis list which contains three different n-best lists with the corresponding scores of the R2L and L2R models. Each n-best list of the L2R models is passed to each R2L model to integrate both lists into a union relation resulting from the summation of the scores and reordered to obtain the k-best list, which is the final output. This notably improves the precision and F1 score, as shown in Table 2, which increases the F1 by 1.22 and 0.90 in QALB-2014 and QALB-2015, respectively. The impact of joint search in the n-best lists R2L and L2R led the system to improve the accuracy of prefixes and suffixes in the output. This approach facilitated more comprehensive exploration of potential corrections, enabling the system to capture and rectify errors specifically related to prefixes and suffixes.

## Bidirectional decoding optimization

This subsection investigates the impact of the most common NMT bidirectional decoding techniques in GEC compared to bidirectional knowledge distillation.

### Asynchronous bidirectional decoding

*Zhang et al. (2018)* proposed an asynchronous bidirectional decoding method that employs a standard encoder-decoder along with a backward decoder. In this work, the existing L2R decoder was used as a backward decoder and the R2L decoder as a forward decoder. R2L decoder generates the correction from right to left, considering the bidirectional source and reversed hidden states of the backward decoder to improve the correction accuracy. Asynchronous bidirectional decoding achieved $F_1$ scores of 68.83 and 71.14 for QALB-2014 and QALB-2015, respectively, as shown in Table 3.

**Table 3** Performance of asynchronous and synchronous decoding in AraGEC using the same baseline (Transformer) compared to BKDGEC, bold highlighting the highest scores.

| Model | QALB-2014 | | | QALB-2015 | | |
|---|---|---|---|---|---|---|
| | Prec. | Recall | $F_1$ | Prec. | Recall | $F_1$ |
| Asynchronous bidirectional decoding | 77.34 | 62.02 | 68.83 | 75.61 | 67.18 | 71.14 |
| Synchronous bidirectional decoding | 77.59 | 62.48 | 69.22 | 75.67 | 67.89 | 71.56 |
| BKDGEC | **77.91** | **63.11** | **69.73** | **76.17** | **68.42** | **72.08** |

### Synchronous bidirectional decoding

To circumvent the limitation of bidirectional decoding, *Zhou, Zhang & Zong (2019)* proposed to integrate the R2L and L2R decoders into a synchronous and bidirectional framework instead of performing independent bidirectional decoding. The same technique has been applied, which used a single decoder to generate the output correction R2L and L2R in an interactive and simultaneous decoding process. The simultaneous decoding achieved 69.22 and 71.56 F1 scores in the QALB-2014 and QALB-2015, respectively, as shown in Table 3. This technique allows the GEC framework to take advantage of the history (backward decoding) and future (backward decoding) information into an interactive decoding process that uses R2L and L2R at the same time.

Bidirectional knowledge distillation differs from the above methods, allowing the system to utilize richer target-side contexts for corrections. This occurs when L2R target-side context and R2L corrections are integrated into an end-to-end joint framework and take the agreement between decoders as a regulation term. Hence, it will much alleviate the error propagation of the reverse target-side context. In summary, Table 3 shows that our bid-knowledge distillation without fine-tuning and re-ranking achieved the best improvement over both methods.

### BLEU score

In this subsection, we assess the performance of the proposed framework using the BLEU score to compare the quality of its outputs to the reference or golden sentences, as well as to the baseline performance (Transformer-based), which is an extra human evaluation. Initially, source sentences of the benchmark QALB-2015 were grouped into eight different lengths, also different settings have been used including n-grams with n from 1 to 4.

Table 4 shows that the proposed model achieved the highest scores in different lengths compared to the baseline trained using the same dataset and hyperparameters. The performance of both models gradually increased with the sentence length, and BLEU score settings changed, with our model being superior as shown in Fig. 4. Once again, this demonstrates the efficiency of the BKDGEC for low-resource GEC systems, which leads to overcome the challenge of exposure bias problem and improved performance without the need for extra resources or training additional models.

The proposed AraGEC framework has been compared with the existing approaches including MT-based, NMT-based, and hybrid systems as shown in Table 5. CLMB-1 is the

**Table 4 Performance of our AraGEC framework was assessed across various BLEU score thresholds and input lengths, with the best scores highlighted in bold.**

| Sentence lengths in words | Unigram | | Bigram | | Trigram | | Fourgram | |
|---|---|---|---|---|---|---|---|---|
| | Transf. | BKDGEC | Transf. | BKDGEC | Transf. | BKDGEC | Transf. | BKDGEC |
| 1 to 29 | 51.65 | **60.43** | 49.80 | **56.14** | 50.14 | **61.11** | 41.20 | **53.11** |
| 30–35 | 57.34 | **67.13** | 56.73 | **62.63** | 57.32 | **69.13** | 59.6 | **68.31** |
| 36–45 | 66.41 | **75.63** | 65.18 | **71.13** | 65.18 | **79.20** | 65.42 | **74.18** |
| 46–55 | 76.84 | **85.31** | 75.42 | **80.03** | 75.61 | **82.19** | 70.09 | **79.64** |
| 56–65 | 80.11 | **87.60** | 78.92 | **86.21** | 79.49 | **86.84** | 75.53 | **82.47** |
| 66–75 | 82.42 | **88.79** | 81.96 | **88.39** | 81.71 | **88.74** | 78.92 | **88.73** |
| 76–85 | 82.21 | **90.63** | 82.13 | **90.72** | 82.39 | **91.35** | 81.02 | **91.23** |
| >85 | 83.14 | **91.23** | 83.22 | **91.72** | 84.52 | **93.64** | 86.32 | **95.14** |

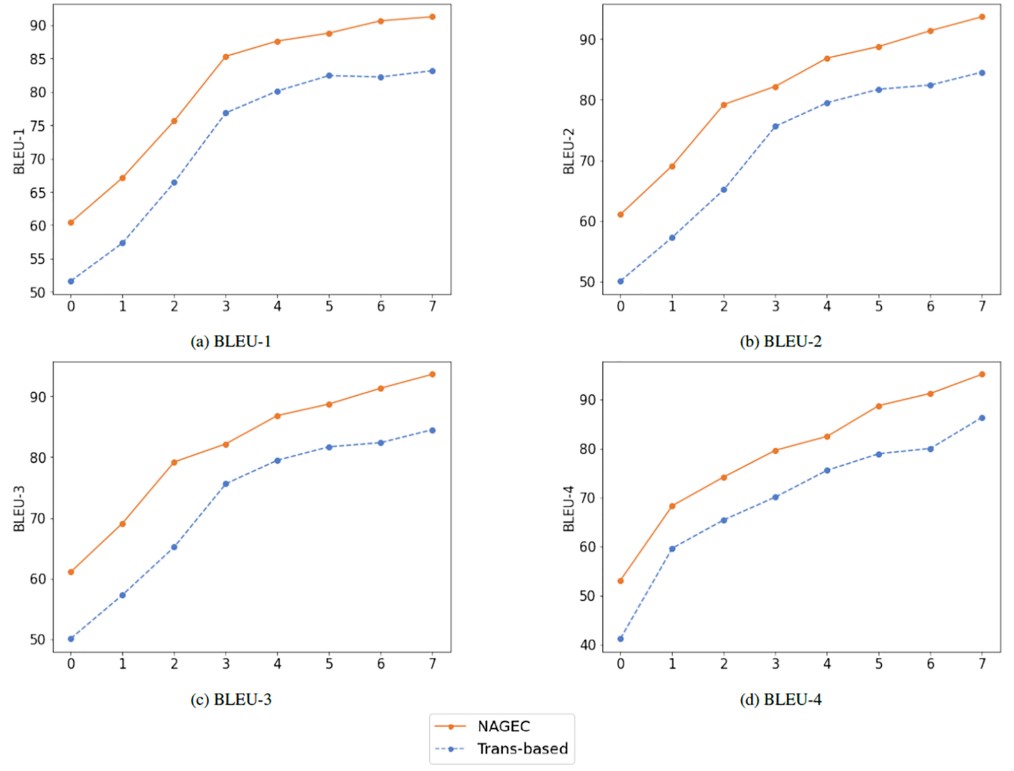

**Figure 4 Performance achieved using different settings of BLEU score.**

best system in the first Arabic shared-task (*Mohit et al., 2014*) which is a hybrid system of machine-learning techniques and linguistic knowledge. SCUT is a neural-based model that employed CNN and attention mechanism. CUFE is a systematic rule-based system for Arabic text correction that achieved the best score in the second shared-task (*Rozovskaya et al., 2015*) of Arabic GEC. AHMADI and WATSON are neural-based models that exploit

**Table 5 Comparisons of $F_1$ scores: AraGEC framework *vs* existing approaches on two benchmarks, bold highlighting the highest scores.**

| Models | 2014 | 2015 |
|---|---|---|
| CLMB-1 *Rozovskaya et al. (2014)* | 67.91 | N/A |
| SCUT *Solyman et al. (2021)* | N/A | 70.91 |
| CUFE *Nawar (2015)* | N/A | 72.87 |
| AHMADI *Sina (2017)* | 50.34 | N/A |
| WATSON *Watson, Zalmout & Habash (2018)* | 70.39 | 73.19 |
| PAJAK *Pajak & Pajak (2022)* | N/A | 69.81 |
| BKDGEC (Our model) | **71.51** | **74.03** |

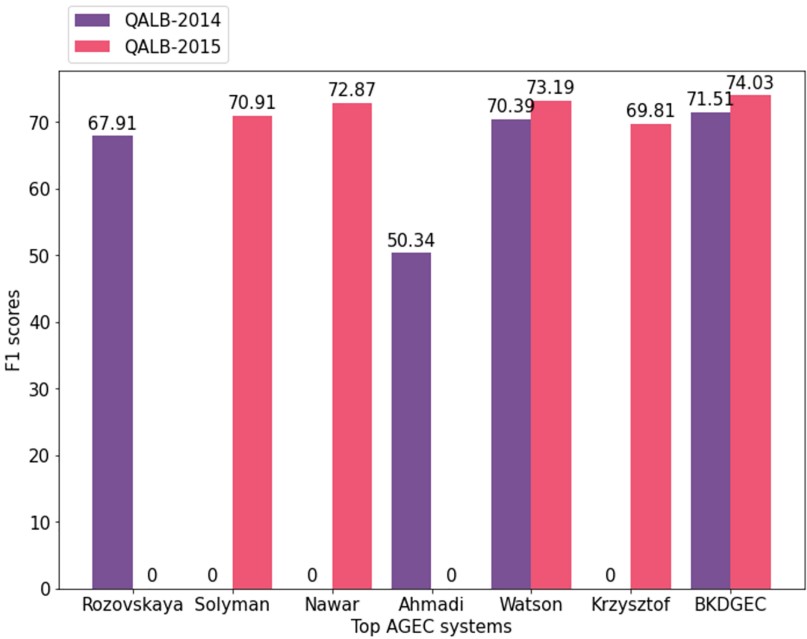

**Figure 5 Visualized $F_1$ scores of leading systems in AraGEC across QALB-2014 and QALB-2015 benchmarks.**

bidirectional RNN in different settings such as *Fasttext* pre-trained embeddings. PAJAK is a multi-lingual neural-based model tuned for GEC. In closing, BKDGEC achieves significant improvements over all AraGEC baselines in two benchmarks as shown in Fig. 5.

## Case study

In this subsection, we assess the performance of different versions of the GEC framework using a real-world example. The example is sourced from the QALB-2015 test set and encompasses a total of 24 errors. Specifically, there are 18 instances of spelling errors (labeled as "sp"), five instances of syntax errors (labeled as "sy") in error number five, and 13 instances of syntax errors (labeled as "sy") in error number 13. Additionally, there are punctuation errors in four instances (labeled as "pt"), six instances (labeled as "pt"), 16 instances (labeled as "pt"), and four instances (labeled as "pt"). Figure 6 shows the output

| Type | Example |
|---|---|
| Source | الصحافه عندنا فى السودان وحتى ²(sp) ¹(sp) الان تفتقد ³(sp) للمصداقيه وتعتمد فى نجاحها واستمراريتها على التطبيق ⁴(pt)* و النفاق ⁶(pt)* والانحياز ⁷(sp) الى ⁸(sp) جماعه ⁹(sp) او ¹⁰(sp) فئه ¹¹(sp) معينه من ¹²(sp) اجل تلميها فقط ⁵(sy) و ليس من ¹⁴(sp) اجل ¹⁵(sp) الصحافه ¹⁶(pt)* * ¹⁷(sp) فبلدى ¹⁸(sp) تحتاج ¹⁹(sp) الى ²⁰(sp) اقلام ¹³(sp) حره ²²(sp) شريفه من ²³(sp) اجل ذلك ²⁴(pt)* ²¹(sp) |
| Target | الصحافة عندنا فى السودان وحتى الآن تفتقد للمصداقية وتعتمد فى نجاحها واستمراريتها على التطبيل ، والنفاق ، والانحياز إلى جماعة أو فئة معينة من أجل تلميها فقط وليس من أجل الصحافة ، فبلدي يحتاج إلى أقلام حرة شريفة من أجل ذلك . |
| English | The press we have in Sudan up to now lacks credibility and it depends on hypocrisy and polishing up a particular group for its success and continued existence, not for the sake of the press. For that, my country needs free and honest writers. |
| Baseline (Transformer) | الصحافة عندنا فى السودان وحتى ²(sp) الان تفتقد ³(sp) للمصداقيه وتعتمد فى نجاحها واستمراريتها على التطبيل ⁴(pt)* والنفاق ⁶(pt)* والانحياز ⁷(sp) الى ⁸(sp) جماعه ⁹(sp) أو فئة معينة من أجل تلميها فقط وليس من أجل الصحافة ، ¹⁷(sp) فبلدى ¹⁸(sp) تحتاج إلى أقلام حرة ²²(sp) شريفه من أجل ذلك . |
| Baseline + EDSE data | الصحافة عندنا فى السودان وحتى ²(sp) الان تفتقد للمصداقية وتعتمد فى نجاحها واستمراريتها على التطبيل ⁴(pt)* والنفاق ⁶(pt)* والانحياز إلى جماعة أو فئة معينة من أجل تلميها فقط وليس من أجل الصحافة . new ¹⁷(sp) فبلدى ¹⁸(sp) تحتاج إلى أقلام حرة شريفة من أجل ذلك . |
| BKDGEC + EDSE data | الصحافة عندنا فى السودان وحتى الآن تفتقد للمصداقية وتعتمد فى نجاحها واستمراريتها على التطبيل ، والنفاق ، والانحياز إلى جماعة أو فئة معينة من أجل تلميها فقط وليس من أجل الصحافة . new ¹⁷(sp) فبلدى ¹⁸(sp) تحتاج إلى أقلام حرة شريفة من أجل ذلك . |
| BKDGEC + EDSE data + Fine-tuning | الصحافة عندنا فى السودان وحتى الآن تفتقد للمصداقية وتعتمد فى نجاحها واستمراريتها على التطبيل ، والنفاق ، والانحياز إلى جماعة أو فئة معينة من أجل تلميها فقط وليس من أجل الصحافة . new فبلدي يحتاج إلى أقلام حرة شريفة من أجل ذلك . |

**Figure 6** Examples of output from different versions of BKDGEC framework, incorrect words are colored in red.

of the baseline, the baseline trained using EDSE data, the proposed model BKDGEC, and BKDGEC with fine-tuning. Furthermore, we provide the source, target, and English translation. Initially, the baseline that was trained using small training data of QALB-2014, corrected fifteen errors and failed to correct seven spelling and two punctuation errors.

Whereas a version of the baseline trained using our data EDSE has successfully corrected most of the reported errors except for three spelling and punctuation errors in 5 and 6 (pt) and caused a new punctuation error labeled "new". BKDGEC model corrected all the errors except two spelling errors in 17(sp), 18(sp), and the new punctuation error. BKDGEC with fine-tuning has made significant improvements and successfully corrected all reported errors, with the exception of the punctuation error labeled as "new".

This indicates that BKDGEC has been somewhat successful in challenging the scarcity of training data and also address the exposure bias problem. However, it still falls short of perfection as it is unable to correct certain punctuation, dialectal words, and challenging grammatical errors when the output of the test set is evaluated on a sentence-by-sentence basis. Therefore, extra effort is needed to correct the dialectal words, punctuation, and the

most complex grammatical errors, taking into account the variability of punctuation usage in Arabic language due to writer's style.

## CONCLUSION AND FUTURE WORK

This article introduced an AraGEC framework based on the Transformer-based equipped with bidirectional knowledge distillation to overcome the exposure bias problem. Furthermore, the proposed model applied a process of knowledge distillation using a Kullback-Leibler divergence method as a regularization term to incorporate the learning information from the backward decoder to the forward decoder. To address the challenge of sparse data in GEC, a novel approach was proposed that utilized a supervised confusion function called the equal distribution technique for syntactic errors, which is used to construct massive synthetic data. The generated data exhibits more training patterns, surpassing the classical confusion methods, and comprises a substantial training set of 25.162 million examples, making it the largest AraGEC training data available. Experimental results on two benchmarks demonstrated that the synthetic data makes a significant improvement, which reported the highest $F_1$ score over the previous AraGEC systems.

In the future, we aim to investigate the influence of the confusion method in producing trustworthy syntactic training data for low-resource languages like Italian, Russian, and Indonesian. In the same context, we are also interested in investigating the impact of bidirectional knowledge distillation on other sequence-to-sequence tasks, such as text classification, image captioning, and conversational models.

## ACKNOWLEDGEMENTS

The authors would like to express deep gratitude to the Chinese Scholarship Council (CSC) and the staff of the School of Computer Science, Wuhan University of Technology for their assistance at every stage of the project and for their invaluable support throughout the entire duration of this project. The authors extend their appreciation to the Deanship of Scientific Research at King Khalid University for funding this work through a large Groups Research Project under grant number (RGP.2/175/44).

### Funding

This work was funded by the Deanship of Scientific Research at King Khalid University through large group Research Project under grant number (RGP.2/175/44). The funders had no role in study design, data collection and analysis, decision to publish, or preparation of the manuscript.

### Grant Disclosures

The following grant information was disclosed by the authors:
King Khalid University: RGP.2/175/44.

## Competing Interests

The authors declare that they have no competing interests.

## Author Contributions

- Zeinab Mahmoud conceived and designed the experiments, performed the computation work, prepared figures and/or tables, authored or reviewed drafts of the article, and approved the final draft.
- Chunlin Li analyzed the data, prepared figures and/or tables, authored or reviewed drafts of the article, and approved the final draft.
- Marco Zappatore conceived and designed the experiments, prepared figures and/or tables, authored or reviewed drafts of the article, and approved the final draft.
- Aiman Solyman conceived and designed the experiments, performed the experiments, performed the computation work, prepared figures and/or tables, authored or reviewed drafts of the article, and approved the final draft.
- Ali Alfatemi conceived and designed the experiments, performed the experiments, prepared figures and/or tables, authored or reviewed drafts of the article, and approved the final draft.
- Ashraf Osman Ibrahim performed the experiments, prepared figures and/or tables, authored or reviewed drafts of the article, and approved the final draft.
- Abdelzahir Abdelmaboud performed the experiments, prepared figures and/or tables, authored or reviewed drafts of the article, and approved the final draft.

## Data Availability

The data, models, and scripts are available at Zenodo: Zainab Obied. (2023). Zainabobied/SLBDEGC: GEC (GEC). Zenodo. DOI 10.5281/zenodo.8108476.

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
