# Peer review of "Semi-supervised learning and bidirectional decoding for effective grammar correction in low-resource scenarios"

_PeerJ Computer Science, doi:10.7717/peerj-cs.1639_

## Round 0.1 · original submission · Major Revisions

The reviewers found the paper to be sound but not ready for publication at this stage to be published. Please address all their concerns.

Reviewer 1 ·

Basic reporting

1. Considering that paragraph (Line 98) talks about low-resource languages, I am not sure why Ge et al. (2018) is placed here since their work is only experimented on English.

2. Line 126, FasTest should be FastText.

3. The process of the noise method is somewhat unclear. Can one sentence include more than one error type? It would be better to use an algorithm to explain the noise method.

In 179, the paper states that N refers to the number of words to be processed and has an initial value between 0 and 1. This also makes me feel confused since the number of words should be an integer. I think that the paper means the ratio.

4. In Line 296, 2024 should be 2014.

5. The information provided in Figure 3 and Table 1 are totally the same; one of them should be deleted.

6. In Line 550, the conference name is not completed; it should be Proceedings of the 2018 Conference on Empirical Methods in Natural Language Processing.

Experimental design

1. The experimental design of synthetic data is somewhat unfair because the data size is different. Previous works (SCF, SSNM) only used 250k examples, and the proposed method used 25 million examples. It would be better to apply the SCF and SSNM to these 25 million examples.

Validity of the findings

no comment

·

Basic reporting

Overall, I think this is a strong paper, especially with regard to the model design and experiments. The language is clear in most areas, with a few specific exceptions (see annotated PDF for notes). The paper could stand to be edited more thoroughly; there are a few typos which may result in confusion by readers. The article is well-structured and the diagrams are clear, with one exception noted in 'additional comments'.

The data and shared in the author's GitHub repository seems incomplete. The authors do not share their system outputs for inspection, and the full code for training and running their models is not yet available. The authors state in the repo that "the whole code files will be released soon!" I am not sure how complete the data and code is expected to be during the review process, but at this time it is incomplete.

The paper is self-contained and has appropriate hypotheses and related experimental results.

Experimental design

The overall experimental design is sound. The authors do several things in this paper. First, they develop a new method of noising Arabic text to be used as pre-training data for an Arabic GEC model. I think that they make a bit of an over-exuberant claim in stating that their noising method covers "most types of errors in the Arabic language." In reality, they only use six rule-based noise injection methods, and they do not provide any examination of error distributions in native text to demonstrate that these methods are representative of the types of errors seen in real Arabic writing. I would encourage the authors to soften these claims, as their approach is perfectly valid, just probably not a true representation of errors (either in type or distribution) of real data. Of course, if the authors did conduct an error analysis of real data, they should include it to show how their approach approximates real-world errors, and that it covers "all types of errors in AraGEC" as they state. As I said, though, I'm not calling the proposed approach into question. Even simple spelling error injection has shown benefits for GEC pretraining, so the authors approach is likely an improvement on simpler methods from previous work.

The authors modeling approach is likewise sound and well-described. I have a bit of issue with the terminology used - "knowledge distillation" usually refers to transferring information from a larger model into a smaller model. In this case, the models are the same, or similar, size. I would also like more information on how the authors settled on their parameter settings for their final models. Finally, I think that the authors statement that they post-process their output may need to be reevaluated; it would be more fair to compare the original model output to previous systems (unless of course such post-processing was used in previous AraGEC work).

Validity of the findings

The authors findings appear sound and are a solid improvement over the previous work to which they compare their results. The authors conclusions are sound given their results.

It would be helpful if the authors provided their system outputs in the GitHub repo so reviewers could inspect the model output themselves.

Additional comments

Figure 3 and Table 1 appear to be reporting the same results. I don't think you need both.

---

## Round 0.2 · accepted · Accept

The paper is ready for publication.

Reviewer 1 ·

Basic reporting

The paper has been greatly improved and can be published in my opinion. Some final checks are needed, Figure 6 is too large,

Experimental design

no comment

Validity of the findings

no comment

·

Basic reporting

This is a revision review. The authors have substantially rewritten their text and have resolved all of my previously identified concerns. Most importantly, the authors have revised their claims relative to the types of errors their approach covers, and have made their code available to reviewers.

Experimental design

As I previously stated, the experimental design is sound. I have no specific issues with the experimental design at this point, given the authors' revisions.

Validity of the findings

The findings seem valid and make sense given related work in more resource-rich languages. Again, I am satisfied that the authors have addressed my primary concerns with their paper, which were never related to the validity of the finding, but to the breadth of the claims.

Additional comments

This is a good paper. A small bit of editing may be needed, but overall the I believe the paper is sound and well-argued.